# Environmental and Occupational Exposure to Asbestos as a Result of Consumption and Use in Poland

**DOI:** 10.3390/ijerph16142611

**Published:** 2019-07-22

**Authors:** Małgorzata Krówczyńska, Ewa Wilk

**Affiliations:** Department of Geoinformatics, Cartography and Remote Sensing, Chair of Geomatics and Information Systems, Faculty of Geography and Regional Studies, University of Warsaw, 00-927 Warsaw, Poland

**Keywords:** asbestos, asbestos-related diseases in counties, environmental exposure, occupational exposure, spatial distribution

## Abstract

Asbestos is harmful to human health; exposure to asbestos causes a wide range of asbestos-related diseases. Aim: Malignant mesothelioma (MM) is unique to occupational and environmental asbestos exposure. Methods: Environmental asbestos exposure was examined in relation to asbestos use and manufacturing, the quantity of the asbestos-containing products still in use, the concentrations of asbestos fibres in the air and the number of MM cases diagnosed each year per county. Results: The correlation coefficient of the measurements of the asbestos fibre concentrations in the air and the quantity of asbestos-cement products in use is high and amounts to 0.68. Meanwhile, the correlation coefficient of the measurements of asbestos fibre concentrations in air and MM morbidity rate resulting from environmental exposure calculated for particular counties in provinces is low and amounts to 0.37. The highest MM morbidity rate was observed for Małopolskie and Śląskie, a typical industrial area of Poland. Conclusions: There are MM cases which are still attributable to occupational asbestos exposure, although MM cases resulting from environmental exposure to asbestos have an increased MM risk. Poland is among those countries with a low MM incidence rate, which seems to be an underestimation of environmental asbestos exposure. As long as asbestos-cement products are used in the environment, actions should be undertaken to protect public health.

## 1. Introduction

The term “asbestos” refers to a group of naturally occurring fibrous serpentine or amphibole minerals [1]. Asbestos was broadly used in industrial production due to its extraordinary tensile strength, poor heat conduction, and resistance to chemical attack [2]. In 1955–1998, over 2.2 million tons of asbestos fibres were imported into Poland [3], including over 60,000 tons of asbestos per year in the years 1970–1990 [4,5,6,7,8,9,10,11,12,13,14,15,16,17,18]. Chrysotile was used mainly in the asbestos-cement construction industry, and asbestos-containing products were manufactured in 28 plants, of which 10 produced asbestos-cement products [19]. Asbestos-cement flat and corrugated sheets were used for roof coverings; pressed, flat, cladding, and panels were used as facades of multi-family buildings [20]. During the period from 1950 to 1998, 1391 million m^2^ of asbestos-cement products were manufactured in Poland, i.e., approximately 15.3 million tons. The peak production period was in the 1970s, when more than 50 million m^2^ of corrugated and flat sheets used in construction were produced annually [19].

The first mention of an adverse effect of asbestos was reported in 1906, and the results of the comprehensive study of the health effects of asbestos were published in 1928 [21]. The World Health Organization (WHO) has pointed out that asbestos is carcinogenic to humans [22]. Exposure to asbestos causes a wide range of diseases, such as asbestosis, as well as cancers such as malignant mesothelioma (MM) and lung cancer [23]. MM is a rare, aggressive form of cancer that arises from mesothelial cells of the pleura, peritoneum, and pericardium, with a median survival of 1 year from diagnosis by asbestos or other asbestiform fibres [24]. The number of cases of asbestos-related diseases depends on the type of asbestos used, and it rises with the increase of the use of crocidolite in the asbestos production [25,26]. The fact that the etiology of individual patients is difficult to determine is because the latency period between exposure to asbestos fibres and the appearance of disease symptoms is long. In the vast majority of cases, more than 20 years pass from the exposure to asbestos to the occurrence of MM [27,28,29]. Lung cancer is detected five times more often than in the general population in people who are exposed to asbestos fibres, and their latency period is around 20–35 years [30].

The occupational exposure to asbestos mainly relates to work on the extraction of asbestos in mines or with the production of asbestos-containing products, as well as during the dismantling, repair and maintenance of the products used and is widely discussed in the literature, particularly since 1960 when mesothelioma and asbestos were causally linked [21,31,32,33,34]. The association between environmental exposure to asbestos and MM has been investigated in populations living in areas not affected by specific environmental sources of asbestos exposure [32]. There were no asbestos mines in Poland. The other sources of environmental exposure are consumer goods still in use with the admixture of asbestos, construction and demolition sites, where products containing asbestos were used, and asbestos in buildings, which formed a part of the elevation, roofing, walls, etc. [35].

Due to the pathogenic nature of asbestos, in 1997, a statutory ban on the production, the use and the marketing of products containing asbestos was introduced in Poland [36]. Considering the long latency period of MM, a high number of cases is still expected in Poland in the next few decades. All products which are now in use shall be removed by the end of 2032. The demolition of asbestos-cement products will constitute a potential risk of the increase of the MM incidence rate. Since in Poland the vast majority of asbestos-containing products are asbestos-cement building materials [37], they formed the basis for the estimation of the potential risk of the environmental asbestos exposure in the undertaken research. Monitoring of the risk assessment of asbestos-related diseases requires detailed data on the quantity and the place of use of asbestos-cement products [38] and the level of concentrations of asbestos fibres in the air [39]. There is no safe threshold of exposure to asbestos; any degree of contact may involve potential risk. The degree of potential risk is related to the exposure. The experience of many countries suggests that attempts to reduce exposure without a concurrent reduction in overall use are insufficient to control risk [40]. The main aim of the undertaken study is to present the current state of asbestos exposure, both occupational and environmental, in Poland with the reference to the consumption, the manufacturing and the use of asbestos-containing products. The previous study undertaken by us was dedicated to stratifying the raw MM morbidity rate by province, which is the highest level of the administrative division of Poland [41]. This paper aims to present the current situation on possible sources of asbestos environmental exposure in counties, which are the lowest level of the administrative division in Poland. It is of high importance to analyse units as small as possible in order to prepare strategies for monitoring the risk of environmental asbestos exposure, since the investigation on the current MM raw morbidity rate in provinces has shown that a more detailed survey is desirable [41].

## 2. Materials and Methods

We have used data on imports of asbestos fibres derived from the United States Geological Survey [2,42,43] and the Statistical Yearbook of Foreign Trade for Poland gathered for the period of 1955 to 1997 [4]. Data on asbestos use and manufacturing plants were collected during the literature review and the field survey undertaken to establish the current state of asbestos manufacturing plants in Poland [20]. The results of the estimation of the amount of asbestos-cement products in Polish provinces obtained by Wilk et al. [3] were used to estimate the amount of asbestos-cement products. All available data on MM cases were derived from the National Cancer Registry, which contains data on MM cases in accordance with the ICD-10 classification under the code of C45, broken down by gender, for the period of 1999–2013 [44]. The research period is related to the introduction in 1994 of the Tenth Revision of the International Statistical Classification of Diseases and Related Health Problems (ICD). In Poland, this revision was enforced in 1996. Measurements of the asbestos fibre concentrations in the air were carried out from 2004–2013 for the Ministry of Economy [41]. Data collected on the amount of the asbestos-cement products and the results of the measurements of asbestos fibre concentrations in the air have then been aggregated to the county level in order to determine the impact of the environmental exposure to asbestos. A structured database was developed based on data gathered and constituted the input for further analysis.

## 3. Results

### 3.1. Asbestos Consumption and Manufacturing

There are no asbestos deposits in Poland that are to be extracted on an industrial scale [45]. In total, in 1955–1998, over 2.2 million tons of asbestos were imported to Poland [4,5,6,7,8,9,10,11,12,13,14,15,16,17,18]. The construction industry was the largest recipient of asbestos fibres in Poland and accounted for about 80–90% of all asbestos-containing products manufactured [45,46,47,48,49,50]. Asbestos-cement flat and corrugated sheets were used for roof coverings; pressed, flat, cladding, panels were used as facades of multi-family buildings. Asbestos-cement pipes were used as pressure pipes in water pipes and gravity pipes in sewers [51]. There were 28 manufacturing plants across Poland [19], of which 10 produced asbestos-cement products (Figure 1). Wilk et al. [20] have estimated the quantity of the asbestos-cement products in use based on best Random Forest models to be 738,068,000 m^2^ (8.2 million tons).

### 3.2. MM Cases in Counties

The total number of reported MM cases in the National Cancer Registry [44] among men and women in Poland in the period of 1999 to 2014 amounted to 3348 (Figure 2).

The raw MM morbidity rate above 20 was recorded in seven counties in Śląskie (Mikołów, Zawiercie, Kłobuck, Gliwice, Mysłowice, Siemianowice Śląskie and Ruda Śląska), in four counties in Małopolskie (Dąbrowski, Chrzanów, Olkusz and Tarnów) and one county of each of the following provinces: Lubelskie (Parczew), Mazowieckie (Szydłowiec), Świętokrzyskie (Busko-Zdrój) and Zachodniopomorskie (Świnoujście) (Figure 3).

A total of 138 MM cases were reported resulting from the prophylactic examinations referred to as the “Amiantus” programme, dedicated to former employees of asbestos manufacturing plants. It allowed the Polish Ministry of Health to provide former asbestos-plant workers with periodic medical examinations and free access to medicines. According to the Nofer Institute of Occupational Medicine, a reliable estimate of the number of asbestos-related occupational diseases is provided within the framework of the “Amiantus” programme [52]. During the period of 2000–2013, 7020 former employees of asbestos manufacturing plants were surveyed within the “Amiantus” programme. The program of preventive examinations was mainly covered by men, who accounted for 2/3 of patients, which reflects the proportion of the employees in the asbestos manufacturing industry, where women accounted for about 30% of employees [52]. A total of 15 cases of MM were detected in Dąbrowski county, where the Szczucin Asbestos Manufacturing Plant was operating. Over 70% of crocidolite asbestos imported to Poland was used in this plant for the manufacturing of large-diameter asbestos-cement pipes [51]. A total of size MM cases were reported in Sandomierz, where, in glassworks, chrysotile was used in the production of flat glass. MM cases were also reported among employees, who worked in asbestos-cement roofing in Gniezno, Radom, Ogrodzieniec, Chrzanów, Ostrów, and Lublin, as well as in Kamienna Góra (brakes and seals) and Piaseczno (asbestos cartons). The remaining number of cases relates to environmental and para-occupational asbestos exposure (Figure 4).

In Southern Poland, in Małopolskie in Dąbrowski county, the total number of registered MM cases amounts to 158. In this county, the Szczucin Asbestos Manufacturing Plant was located, where crocidolite was used for production, and asbestos-cement waste was also used for 30 years in the Szczucin community for road hardening, courtyards and sports facilities [53], which had a significant impact on the number of diagnosed MM cases related to environmental exposure. The next county with the highest morbidity rate is Busko-Zdrój in Świętokrzyskie in Central Poland, where no asbestos plant was located, but a large quantity of asbestos-cement roofing was used. Until now, the amount of products used in this county is one of the highest in Poland and amounts to almost 500 kg of asbestos-cement products per capita. In Śląskie, the lowest value of MM morbidity rate is 4.62, whereas, in 24 out of 36 counties, the MM morbidity rate is higher than 10. A similar situation is observed in Małopolskie.

### 3.3. Asbestos Fibres Concentrations Measurements in Counties

The analysis of measurements of asbestos fibre concentrations in the air in the period 2004–2013 was made on the basis of data provided by the Ministry of Economy [54]. Measurements of asbestos fibre concentrations were undertaken in 279 counties, which constitute 73% of the total number of counties in Poland (Figure 5).

The highest average concentrations of asbestos fibres in the air were observed in Lubelskie (five counties), Łódzkie (three counties), Małopolskie (three counties), Mazowieckie (one county) and in Warmińsko-Mazurskie (one county). The reported concentration values were above 5000 fibres per m^3^. In 2004, concentrations of over 5000 fibres per m^3^ were recorded in seven counties, and, for the period 2005–2013, in one county. The highest average asbestos fibre concentrations in the air were recorded in 2004 in Łódzkie (in Zgierz, it amounted to 8150 fibres/m^3^) and in Lubelskie (in Kraśnik, it amounted to 8229 fibres/m^3^). In 12 counties, no asbestos fibres were found in the air or a number below the quantification of the method was noted (Figure 6).

### 3.4. Correlation of the Amount of Asbestos-Cement Products in Use in Relation to Asbestos Fibres Measurements and the Number of MM Cases

The correlation coefficient of the measurements of the asbestos fibre concentrations in the air and the quantities of asbestos-cement products in use in counties is high and amounts to 0.68. If Lubuskie is excluded, since there were no measurements undertaken, and Kujawsko-Pomorskie and Zachodniopomorskie, for which the results of measurements in all counties were below the defined level of the method, the correlation coefficient increases to 0.70. Since in 1997 a ban on asbestos was introduced in Poland and the asbestos fibre concentrations in the air measurements were made more than 20 years later, it may be assumed that exposure to asbestos is the result of using asbestos-cement roofs.

The correlation coefficient of the measurements of asbestos fibre concentrations in the air and MM morbidity rate resulting from environmental exposure calculated for counties in provinces is low and amounts to 0.37. Further analysis was performed in counties, including the following (Table 1):-the average rate of the environmental MM morbidity rate (excluding MM occupational cases);-the average amount of asbestos products (in tons);-average number of asbestos fibres concentrations in the air per m^3^;-the average amount of asbestos products in use per person;-the number of plants that used asbestos in production;-percentage share of zones classified into class C, in which the specified criteria values have been exceeded for PM2.5 [55];-percentage share of zones classified to class C, in which the specified criteria values have been exceeded for PM10 [55].

## 4. Discussion

Environmental asbestos exposure is identified in the literature through a questionnaire completed by participants or their proxies or the number of asbestos sources [56]; however, the low quality of proxy responders is increased [57]. It is also underlined that using the registry of MM patients, it is desirable to document the impact of the asbestos environmental exposure [34]. The incidence of MM in the female population is mentioned as having great potential as an indicator of environmental asbestos exposure [58]. Moreover, the information on asbestos manufacturing plants is of great importance for the assessment of the potential influence of environmental exposure to asbestos [56]. As it is desirable to include other explanatory factors of environmental exposure to asbestos [59], results of measurements of asbestos fibre concentrations in the air as the result of the release of asbestos fibres from products were considered.

Since in Poland, all employees of former asbestos manufacturing plants have been included in the “Amiantus” programme, it is possible to determine the number of MM cases in counties through a development of a database gathering and ordering data from different sources, in particular in relation to the number of MM cases due to the occupational exposure and all MM cases, which are collected in the National Cancer Register [44]. It is expected that environmental exposure to asbestos will constitute an increasing factor of MM risk [60]. The widespread low-level environmental exposure to asbestos, most importantly asbestos-cement roofs in buildings, results in a higher number of MM cases [34]. In many countries, measures have been taken to limit or completely prohibit asbestos mine extraction and the production of asbestos-containing products [61], but products containing asbestos are still used, and their influence on asbestos environmental exposure still requires confirmation [62].

The number of registered MM cases in Poland increases each year (Figure 7). In subsequent years, 266 MM cases were reported in 2010, 260 in 2011, 299 in 2012, 326 in 2013 and 297 in 2014.

Compared to Italy and Germany, where the consumption of asbestos products per capita is similar to that in Poland (Poland and Italy 13 kg per capita; Germany 17 kg per capita) (Figure 8), the number of reported MM cases in other European countries are much higher than in Poland.

About 1500 new MM cases are reported annually in Italy due to occupational and environmental exposure. By December 2016, 2635 MM cases were included in the Italian Cancer Registry, referring to the maturity period 1993–2015 [63]. In Poland, during the period of 1999–2014, almost 10-times fewer cases were recorded, in total 3348. For comparison in Germany, the number of reported MM cases was 944 in 2013, 881 in 2014 and 976 in 2016 [64]. This is about three-times more diagnosed MM cases per year in Germany when compared to Poland.

Comparing annual data on MM incidence rates in selected European countries [65] (Figure 9), the highest, and still increasing, MM incidence rate is observed for Great Britain. In Italy, the peak was observed in 2012, and the MM incidence rate is still increasing. In Poland, the MM rate is at the lowest level, and a growing trend is observed.

Environmental asbestos exposure in the general population is related to the occurrence of asbestos fibres in the air. Measurements of asbestos fibre concentrations in the air show relatively low levels of atmospheric air pollution with asbestos fibres, which amount to an average of 400–900 fibres per m^3^ in the area of Western and Northern Poland, and in Southern, Central and Eastern Poland—over 1000 to 2240 fibres per m^3^, which does not exceed the cumulative, normative values. It is worth noting, however, that asbestos concentrations occurring in municipal exposure are sufficient to initiate the development of asbestos-related cancers [66].

The observed increase in MM morbidity rate in the last decade in Poland probably results not only from the higher incidence rate but also from the improvement of diagnostic methods, both in the possibility of collecting material for research and precise pathological evaluation allowing for the proper diagnosis [67,68]. Taking into account the long latency period, environmental asbestos exposure is difficult to study, and it should be evaluated with reference to environmental contamination [34,69]. The risk of MM incidence increases with the air pollution level [41]. The high level of air pollution was determined in a five-year assessment of the air quality in zones in Poland [55]. The areas with the highest degree of air pollution with substances that pose a health risk can be found in Śląskie, Małopolskie and Świętokrzyskie [55]. Due to biological reasons, it is reasonable to assume that different exposures to the same factor have an effect [70]. This may explain the very high rate of disease in Śląskie and Małopolskie, where the air quality assessment revealed levels exceeding the permissible level of PM2.5 and PM10 in the air (Table 1).

## 5. Conclusions

The undertaken study is focused on presenting the asbestos issue in Poland with regard to the underestimation of environmental asbestos exposure. Environmental asbestos exposure was examined in relation to asbestos use and manufacturing, the quantity of asbestos-containing products still in use, the measurements of asbestos fibre concentrations in the air and the number of MM cases diagnosed each year per county (local administrative unit). This may constitute a basis for research on determining spatial patterns of asbestos-related diseases in relation to the number of products used in counties, which are the lowest division in the administrative system of Poland. In-depth research will be undertaken in order to develop a model on the number of cases with the use of a developed database relating to the asbestos fibre concentrations in the air and the amount of asbestos-cement product in use divided into counties. The undertaken study confirms that asbestos pollution is a threat to people living in polluted areas, due to not only the occupational exposure in an industrial environment with the direct use of asbestos but also the presence of asbestos in the environment, in particular, from asbestos-cement roofing. As long as asbestos-cement products are used in the environment, actions should be undertaken to protect public health. Since the extraction, production and use of asbestos products is still taking place in many countries, the problem of the incidence of pleural mesothelioma will increase in subsequent years.

## Figures and Tables

**Figure 1 ijerph-16-02611-f001:**
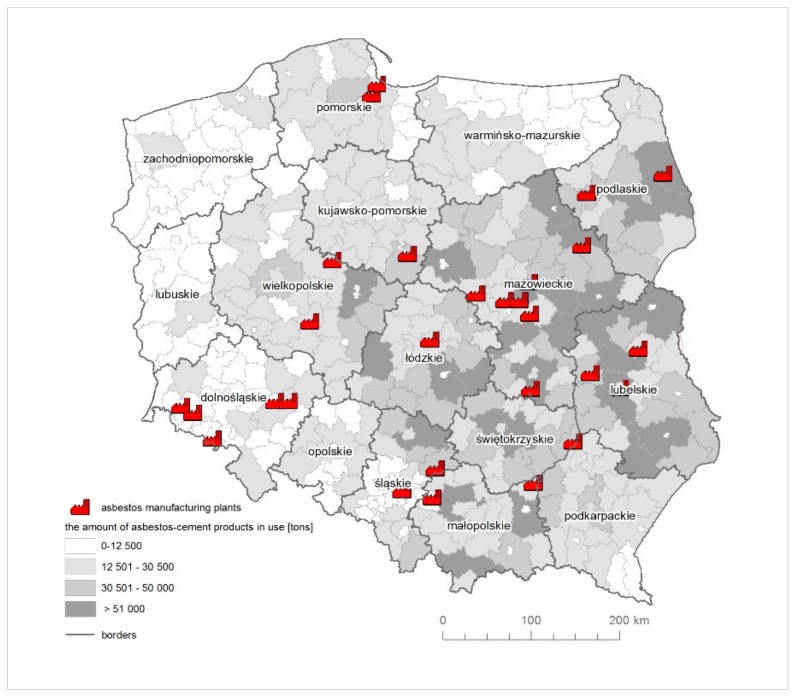
The quantity of asbestos-cement products in use [5] with the reference to the asbestos manufacturing plants.

**Figure 2 ijerph-16-02611-f002:**
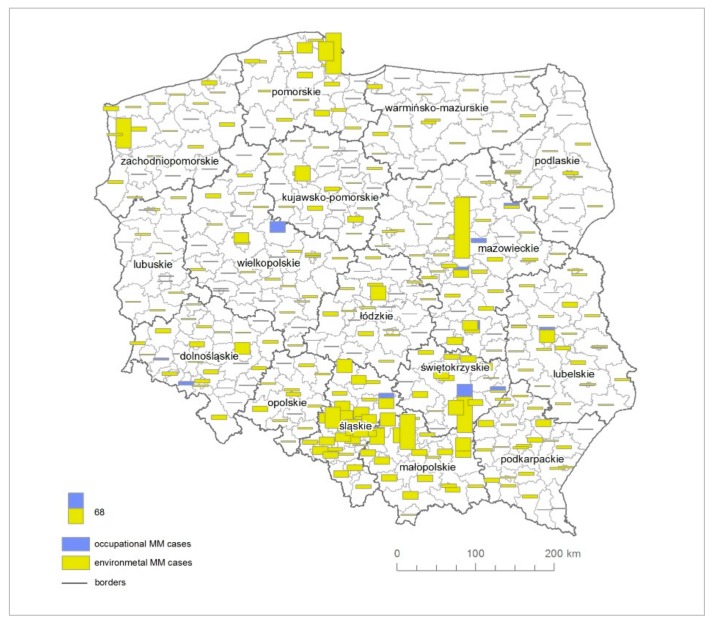
The absolute number of malignant mesothelioma (MM) cases due to the environmental and occupational asbestos exposure.

**Figure 3 ijerph-16-02611-f003:**
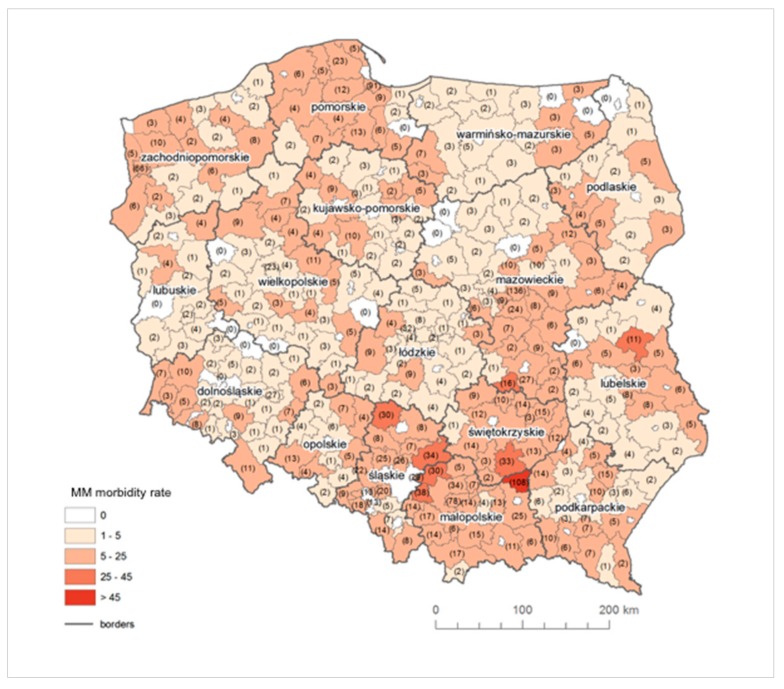
The raw MM morbidity rate in counties (in brackets: the absolute number of MM cases).

**Figure 4 ijerph-16-02611-f004:**
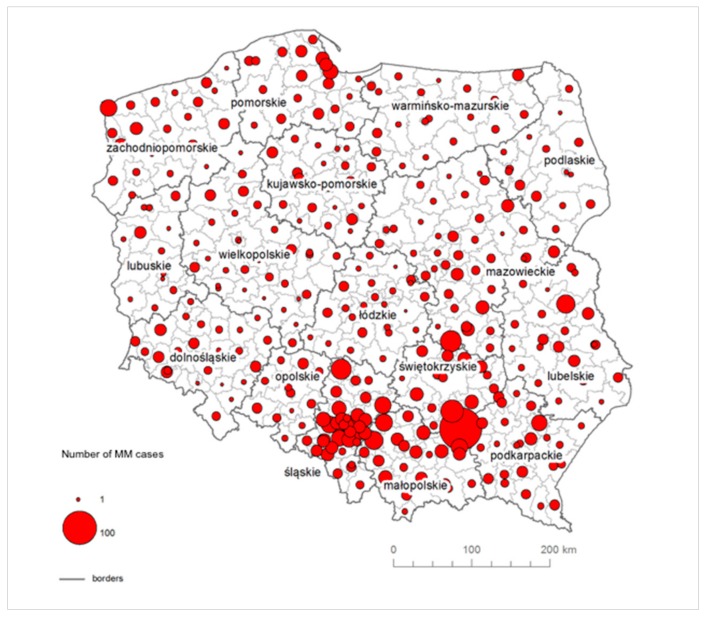
The absolute number of MM cases in counties due to the environmental exposure to asbestos.

**Figure 5 ijerph-16-02611-f005:**
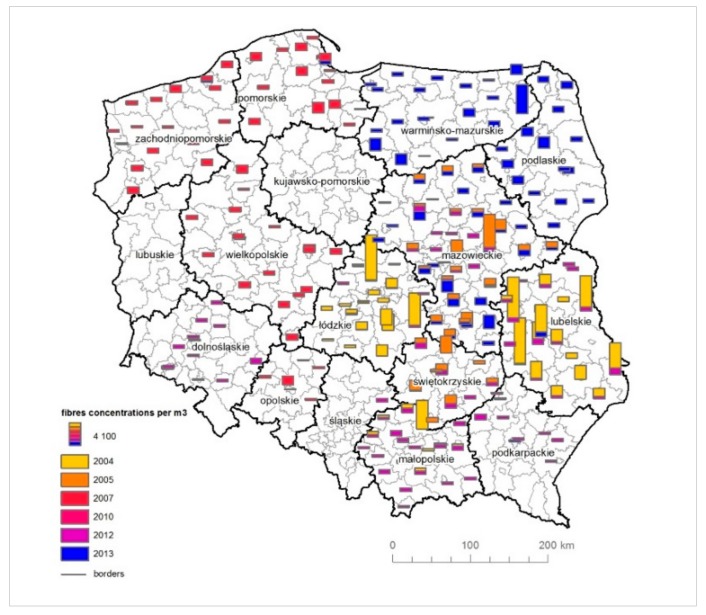
The results of measurements of the asbestos fibre concentrations in the air by counties with regards to the period of 2004–2013 (fibres/m^3^).

**Figure 6 ijerph-16-02611-f006:**
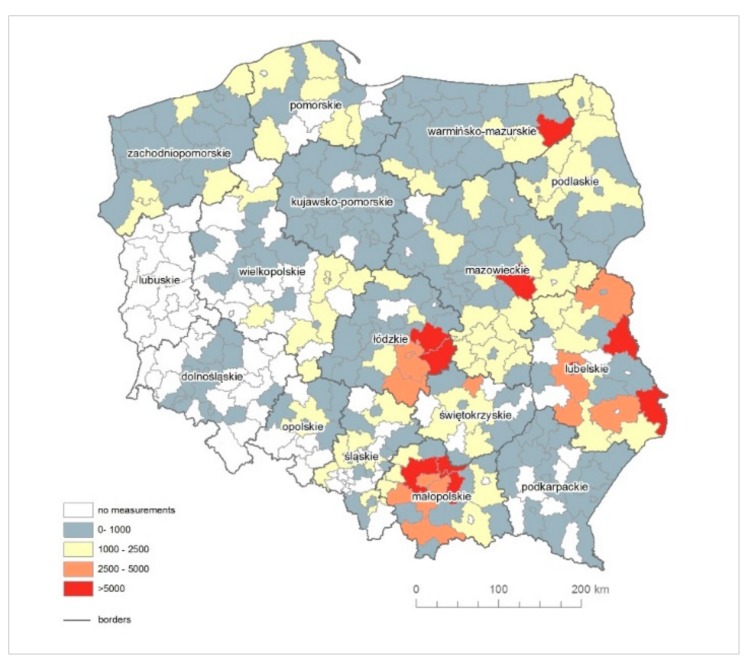
The measurements of the asbestos fibre concentrations in the air by county.

**Figure 7 ijerph-16-02611-f007:**
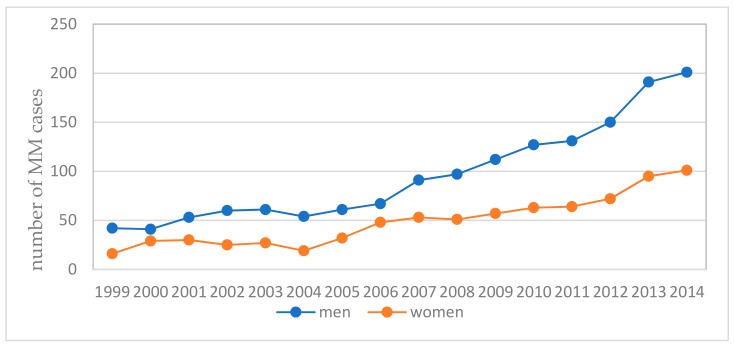
The number of registered MM cases in Poland.

**Figure 8 ijerph-16-02611-f008:**
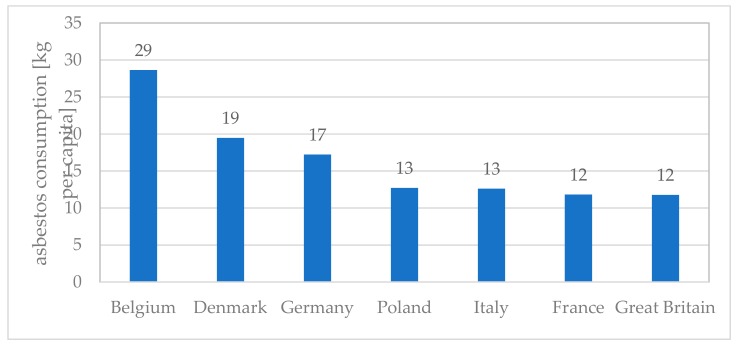
The asbestos consumption per capita in selected European countries.

**Figure 9 ijerph-16-02611-f009:**
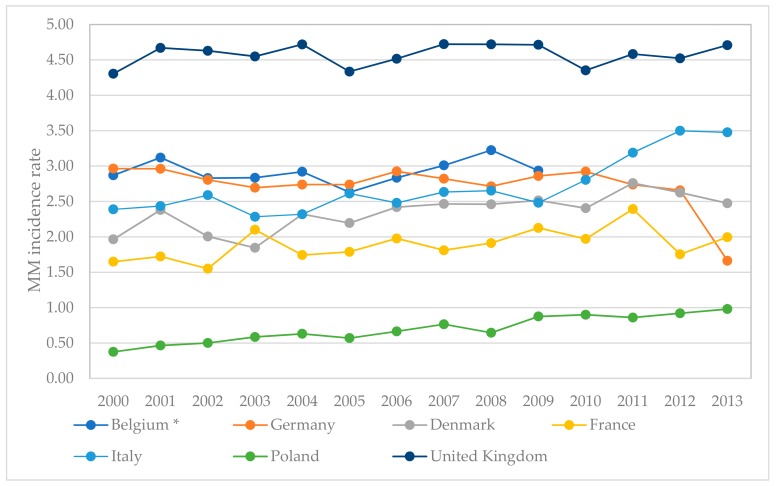
The MM incidence rate in selected European countries.

**Table 1 ijerph-16-02611-t001:** MM morbidity rate in Poland in relation to the amount of asbestos-cement products, asbestos manufacturing plants and the air pollution zones.

Province	MM Morbidity Rate	Average Quantity of Asbestos-Cement Products (tons) [3]	Average Value of the Asbestos Fibres Concentrations in the Air (per m^3^)	Asbestos-Cement Products per Person	Number of Asbestos Manufacturing Plants	% Zone C PM2.5	% Zone C PM10
Dolnośląskie	4.71	9244	507.25	95	5	25%	25%
Kujawsko-Pomorskie	4.88	18,125	556.70	199	1	25%	0%
Lubelskie	6.38	39,842	2710.18	445	3	50%	0%
Lubuskie	3.43	7851	n.m.	108	0	0%	0%
Łódzkie	4.19	32,621	1643.60	313	1	100%	0%
Małopolskie	19.09	25,770	2240.56	168	2	50%	50%
Mazowieckie	6.63	36,272	1111.97	286	7	67%	67%
Opolskie	5.56	14,395	397.02	173	0	50%	0%
Podkarpackie	7.00	20,129	587.24	236	0	0%	0%
Podlaskie	4.05	31,707	1048.21	452	2	50%	0%
Pomorskie	8.27	13,633	904.44	118	2	0%	0%
Śląskie	14.11	12,968	1033.90	102	2	80%	50%
Świętokrzyskie	12.06	37,137	1176.83	412	1	0%	60%
Warmińsko-Mazurskie	4.45	10,304	1135.77	150	0	0%	0%
Wielkopolskie	3.71	19,963	816.98	201	2	67%	0%
Zachodniopomorskie	7.35	8679	777.09	106	0	0%	0%

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
