# Peer review of "Environmental and Occupational Exposure to Asbestos as a Result of Consumption and Use in Poland"

_ijerph, 2019, doi:10.3390/ijerph16142611_

Round 1

Reviewer 1 Report

This report is potentially relevant to the understanding of asbestos use in Poland and the occurrence of the asbestos related disease malignant mesothelioma but is poorly written and adds little to the issues of importance.

It is mainly descriptive and has no significant scientific merit. 

The term morbidity would probably be better replaced by incidence or mortality

The text is poorly written and would benefit from editing by someone with a  better command of the English language so that its relevance could be appreciated

Author Response

Dear Reviewer, 

thank you very much for all your remarks which enabled us to review our manuscript in order to achieve a good quality paper. All your points have been addressed. Please find attached the response to your comments in the attached file.

Yours sincerely, 

Authors

Reviewer 2 Report

Age is an important factor for cancer incidence. Hence, a comparison between regions should be based in general on age-standardized incidence rates.

In figure 4 it is not clear, which kind of ratio is presented here.

My impression is that the demolition of asbestos cement roofs is the real problem, rather than the actual use. Therefore, roofers should have an increased risk.

The authors suspect that the observed increase in MM morbidity rate in the last decade in Poland results not from the actual higher incidence rate, but from the improvement of diagnostic methods, both in the possibility of collecting material for research and precise pathological evaluation allowing for the proper diagnosis.

It would be helpful to compare time trends in the incidence between Poland and the other mentioned countries.

Author Response

(The authors gave the same response as above.)

Round 2

Reviewer 1 Report

The sources of potential exposures in relation to the sampling of fibres are not specified.

The types of asbestos that have been in use have not been analysed 

Reviewer 2 Report

Incidence and mortality rates are usually given per population unit. In your case it seems to be per 100,000 inhabitants. Please insert this in the method section and in tables/figures.